# Women *Quazi* in a Minority Context: An Overview of Sri Lankan Experience

**Mohammad Ismath Ramzy** [1,*] **and Simin Ghavifekr** [2]

[1] Department of Educational Foundations and Humanities, Faculty of Education, University of Malaya, Kuala Lumpur 50603, Malaysia

[2] Department of Educational Management, Planning and Policy, Faculty of Education, University of Malaya, Kuala Lumpur 50603, Malaysia; drsimin@um.edu.my

[*] Correspondence: ismathramzy@um.edu.my

**Abstract:** A woman's eligibility to be appointed as a judge in Shariah courts in Muslim societies has been a debated issue for decades. Although some Muslim majority countries, including Arab countries, have allowed women judges (*Qudath*) in Shariah courts, the Muslim Religious Leadership in Sri Lanka, namely All Ceylon Jamiyathul Ulama (ACJU) is opposed to such appointment to administrate Muslim matrimonial law on the basis of classical Muslim scholars' discussion on the qualification of a judge (*Qadi* in Arabic), particularly referring to their debate on gender; however, women activists in Sri Lanka argue for women *Quazi* on the basis of women's privacy and fair hearing. This article, therefore, explores the Islamic standpoint regarding women *Quazi* in Sri Lanka. Hence, this research studies the classical scholars' discussions on the qualification of a judge (*Qadi*) critically and uses textual and document analysis to bring out the dynamic interpretations of the verses of the Quran and Hadiths that they used for their arguments. The contextual analysis was carried out to understand the various applications of these verses of the Quran and Hadiths in history, particularly in connection with the present situation for women in Sri Lanka. This research found no explicit verses of the Quran and Hadiths to allow or deny women *Quazi.* The positive and negative approach to women judges (*Qudath*) has been founded throughout history on the basis of Islamic scholars' understanding of a few verses of the Quran and Hadith that are related to women leadership. This study recommends women *Quazi* for Sri Lankan *Quazi* courts by highlighting differences of context and insignificance of classical Muslim scholars' debate on gender as a qualification of a judge (*Qadi*).

**Keywords:** women *Quazi*; Sri Lanka quazi court; Muslim personal law; Muslim minority issues; Sri Lankan context of Islam

## 1. Introduction

Appointing female judges (Sing:*Qadi and Pl:Qudath* in Arabic) to administrate *Shariah* law—the law derived from the religious sources mainly the Quran and Hadiths to guide Muslims towards a practical expression of religious conviction—has been a debated issue for decades among Muslim scholars. *Shariah* law is a body of religious and legal scholarship involving the law of *Ibadat*—religious duties, penal law including *hudud*-criminal cases, *Qisas*-retaliation cases and *Ta'zir*—disciplinary cases such as insulting religion, threatening to right or peace etc., transaction law, family law and succession law. Family law is an important part of Shariah.

Muslim scholars in several Muslim majority countries, such as Malaysia, Sudan, Egypt, Tunisia, Yemen, Bangladesh, Pakistan, Indonesia and the Maldives, have recently allowed qualified women to preside over *Shariah* courts [1]. Although Sri Lanka does not have a Shariah court, Sri Lankan Muslims

are allowed to practice marriage and divorce as per Muslim religious matrimonial law. The judicial officer to administrate the Muslim matrimonial law in Sri Lanka under the Muslim Marriage and Divorce Act (MMDA) is called *Quazi*. Since Muslim matrimonial law is considered a part of Shariah law, the appointment of women *Quazi* to administrate the Muslim matrimonial law in Sri Lanka became controversial.

It should be noted that this research differentiates between *Qadi*—Arabic term and *Quazi*-the term used in the Sri Lankan Muslim Marriage and Divorce Act. *Qadi* (in Arabic) is used to denote the judge to administrate Shariah law, while *Quazi* (in Sri Lanka) is used to denote the judge to administrate the Sri Lankan Muslim matrimonial law only.

The debate on the appointment of *Quazi* in Sri Lanka begins with the demand of Muslim women activists for female *Quazi* and the resistance from All Ceylon Jamiyathul Ulama (ACJU), the accepted authority concerning religious affairs of the Sri Lankan Muslims. Women activists mandate the appointment of female *Quazi* on the basis of fair hearings and women's privacy, while ACJU opposes this proposal on the basis of classical scholars' discussion on the qualification of a judge (*Qadi*). ACJU also supported their argument with certain Quranic verses on women leadership such as 4:34, 2:228, 33:35, 66:12, 27:32, 9:71 and 5:42, as well as Hadiths such as "*I have not seen anyone more deficient in intelligence and religion than you*". (Al-Bukhari, Hadith No: 304, 1462, 1951, 2658; Muslim, Hadith No: 214; al-Tirmidhi, Hadith No: 2614; Abu Dawud, Hadith No: 4679; Ibn Maja, Hadith No: 4003).

Even though the appointment of women *Quazi* has been provocative among the Muslim community in Sri Lanka for decades, this has become more intense recently due to ongoing constitutional reform [2]. As far as the constitutional reform includes an amendment of the Muslim Marriage and Divorce Act (MMDA), a decision on *Quazi* appointment [2] is necessary for the completion of the constitutional reform exercise. Although ACJU worked with the MMDA reform committee as a member, disputes over certain issues including women *Quazi* delays constitutional reform exercises [3]. Hence, the arguments on women *Quazi* for and against is continuing and this research intends to contribute to the discussion and to find an agreeable solution.

To the extent that the present debate on women *Quazi* in Sri Lanka is referred to the classical literature on the qualification of a judge (*Qadi*), this research briefly summarizes the classical scholars' discussion on the qualification of a judge (*Qadi*) with a special focus on gender differences. Since there is no explicit *Nas* (the verses of the *Quran* or *Hadith*) on the qualification of a judge (*Qadi*), *fuqaha* (Islamic jurists) and Islamic scholars expounded it from the verses of the *Quran* and *Hadiths* that are related to women leadership. The contribution of Ibn Jarir al-Tabari (AD 835–923), al-Marghlnani (AD 1135–1197), Ibn Quadama (AD 1147–1223), Ibn Farhun (d.1397), Muhammad Sallam Madkur and Muhammad Sangalagi is exceptional as they studied the qualifications of a judge (*Qadi*) from various sources and compiled this in their works. Recently, Ghulam Murtaza Azad compiled these scholars' discussion on the qualification of a judge (*Qadi*) and highlighted that some scholars quantified them to 30, while some others counted only three [4]. For Ibn Qudamah, the judge (*Qadi*) needs only three qualifications: (1) *Kamal* (perfection)—*Kamal al-Ahkam* Mental Perfection, which refers to an adult male who is eligible to observe religious duties, and *Kamal al-Khilqah*—Physical perfection, which refers to a person who does not suffer from physical deficiency; (2) *Adalah*-Justice, which refers to the genuineness of a judge in his sayings, doings and jurisdictions; and (3) *Ijtihad*—the ability to process legal decisions independently on the basis of the primary sources of Islam, which enables him to understand the different opinions and consensus of the previous Muslim jurists [5]. Even though all of the scholars agreed on being a Muslim, matured and learned, they differ in matters such as freedom, being blameless or graceful, as well as gender [6].

### 1.1. Gender Is a Qualification for the Post of a Judge (Qadi)

The classical *fuqaha*, including the representatives of four Sunni *Mazhabs* (schools): Maliki, Hanafi, Shafie and Hanbali, prohibited the appointment of women for the post of a judge (*Qadi*). The Shafie and the Maliki consider gender as a qualification [7]. For instance, being male is one of the qualifications

for being a judge (*Qadi*) in the view of Al-Nawawi (AD 1237–1277) [8]. Similarly, Ibn Farhun (d 1397), a Malikite jurist, prevents women from rising to the post of a judge (*Qadi*) on the basis of gender [9]. Al Sarakhsi (d 899) wrote that Shiah al-Ja'fariyyah and the Zahiri School unanimously agree that a judge (*Qadi*) must be male, not female [5,10]. Ibn Qudamah also considered being female as being disqualified for the post of a judge (*Qadi*) while explaining the qualification of *Kamal*—perfection as mentioned before [5].

In modern times, Syed Jamaluddin al-Afghani (AD.1838–1897), Muhammad Abdul Qadir Abu Faris (b.1956), Al-Shiekh Salam Bin Fahd al-Awdah (b.1955) and Abdul Majid Al-Zindani (b.1942) opposed the appointment of women judges (*Qudath*) [11]. Similarly, Muhammad Abu Faris (AD 1938–2015), Shiekh Atiyyah Saqar (AD 1914–2006), the former president of the Fatwa committee, Al Azhar University [12] and Mustafa Siba'I (AD 1915–1964) [13] are some other prominent scholars who deny women judges (*Qudath*).

*1.2. Gender Is Not a Qualification for the Post of a Judge (Qadi)*

On the other hand, the scholars who support the appointment of women judges (*Qudath*) do not consider gender as a qualification for the post of a judge (*Qadi*). For them, gender does not prevent women from becoming a judge (*Qadi*) insofar as they possess the other qualifications. [11]. Hence, the majority of scholars in the Hanafi School allowed women judges (*Qudath*) to administrate the entire Shariah Law except *Hudud*—criminal and *Qisas*—retaliation cases in Penal Law [11] referring to corporal punishment for these cases [14]. Ibnu Qassim from Maliki's school opined that women could be appointed as a judge (*Qadi*) in all cases except *Hudud* cases [11]. The appointment of women as judges (*Qudath*), for Al Abari, is permissible with the ability to make a legal decision or the ability to give a *fatwa*-decree. Imam Ibn Hazm also permitted the appointment of women judges (*Qudath*) in all affairs of Shariah law [11].

In the modern period, Yusuf al-Qardawi agrees with the opinion that a woman can be appointed as a judge (*Qadi*) except in *Hudud* and *Qisas* cases. He said that there is no clear evidence in *Shariah* to prohibit women from attaining this post [11]. Similarly, Muhammad al-Ghazzali (AD.1917–1996), Abdul Careem Zaidan (AD.1917–2014), Muhammad Baltaji and Tawfeeq al-Wai are some other significant scholars who approved the appointment of women judges (*Qudath*) [11].

The 73rd conference of the National Fatwa Committee Council of Malaysia, which was held on 4 to 6 April, 2006, ruled that a woman could be appointed as a Shariah court judge (*Qadi*) except for *Hudud* and *Qisas* [15]. The State Fatwa Committee of Selangor, Malaysia also endorsed this fatwa during the conference on 26 and 27 June, 2006 [14].

In accordance with the basic principles of *Shariah*, "*all things are originally deemed permissible as long as there is no Shariah text that prohibits them*" [16], meaning that the scholars who allowed women judges (*Qudath*) are not required to provide any evidence to support their view. However, their opponents need to provide evidence to prohibit the appointment of women as judges (*Qudath*).

The scholars who oppose the appointment of women judges (*Qudath*) consider men to be the protectors of women and that women are subordinate to men based on the interpretation verse 4:34 of the *Quran*. This verse says,

> "Men are the protectors and maintainers of women because Allah has made one of them excel over the other, and because they spend out of their possessions (to support them). Thus righteous women are obedient and guard the rights of men in their absence under Allah's protection. As for women of whom you fear rebellion, admonish them, and remain apart from them in beds, and beat them. Then if they obey you, do not seek ways to harm them. Allah is Exalted, Great" (The *Quran*, 4:34).

From this verse, these scholars understood that the guardianship of women is limited to men. This is because men are the financial providers for the household and enjoy ultimate legal rights [6]. They also believed that men are given more intellectual ability, power and patience; therefore, men

have to supervise women and plan her duties and responsibilities [17]. Based on this understanding, these scholars prohibit the appointment of women judges (*Qudath*).

The opponents of women judges (*Qudath*) support their position with several Hadiths that oppose women's superiority over men. For instance, it is reported that the messenger of Allah (pbuh), upon hearing the appointment of Chosroes' daughter as the Persian ruler, said, "*Never shall a folk prosper who have appointed a woman to rule them*" (Al-Bukhari: 4163, Tirmidhi, Fitan, 75: Nas ai, Qudat: 8 and Ahmad v. 43, 51, 38 and 47). Hence, women, from their perspective, should not be appointed to any leadership post including legal affairs in order to ensure the prosperity of the community.

They also cited another Hadith, "*women are weak in reason and religion*" (Al-Bukhari, Hadith No: 304, 1462, 1951, 2658; Muslim, Hadith No: 214; al-Tirmidhi, Hadith No: 2614; Abu Dawud, Hadith No: 4679; Ibn Maja, Hadith No: 4003) to exclude women from the appointment of a judge (*Qadi*). In addition to that, these scholars provide several logical arguments to prove the psychological differences between men and women. Some of them said that judicial administration may cause injury to an unborn child [18], maternity leave will cause interruptions to the court's administration [18] and that menstruation makes females nervous, unstable and leads to sudden changes in their decisions [18]. Hafiz Anwar (2000) has compiled these arguments to disqualify women for the post of a judge (*Qadi*) [19].

In response to these arguments, the supporters of women judges (*Qudath*) showed another aspect of interpretation of the verses of the *Quran* and the Hadiths that were cited to disqualify women to the post of a judge (*Qadi*). In regard to verse 4:34 of the *Quran*, they said that this verse contains instructions and guidance in family disputes instead of subordinating women to men [20]. Therefore, they claim that citing this verse to prohibit the appointment of women judges (*Qudath*) is improper.

Mohammad Fadel (2011) studied the quoted Hadith, "*Never shall a folk prosper who has appointed a woman to rule them*" from the perspective of *Takhsis al-'Amm'*—specifying the general term or Hermeneutical historicism approach and said that this Hadith did not reject female leadership in the judiciary; rather, it vindicates the Prophet Mohammad (pbuh)'s claim of prophecy [21]. Hence, in the view of Fadel, citing this Hadith to refuse to appoint women judges (*Qudath*) is inadequate [22]. Fadel's argument might be further expanded to say that women leadership as judges (*Qadi*) in Islamic legal affairs has only been questioned while Muslim society accepted women leadership in daily life, such as a magistrate in conventional court or as a medical officer to decide treatment processes or a head of institution to manage male staff.

From the above discussion, it is shown that there is no explicit order on the qualification of a judge (*Qadi*) in the *Quran* or Hadith. However, the scholars in classical and modern periods have debated the gender differences in the qualification of a judge (*Qadi*) based on their interpretations and understanding of certain *Quranic* verses and Hadiths related to women leadership. Therefore, this study reviews these interpretations in the Islamic literature in order to perceive an implementation strategy that is fitting to the Sri Lankan context.

## 2. Background

Sri Lanka has a dual legal system regarding marriage and divorce. People have the choice of marrying under either the General law or the specific cultural law. The Sinhalese have the option of marrying under whichever law they like, for example the 1952 Kandyan Marriage and Divorce Act or the 1907 General Marriage Registration Ordinance (GMRO) [22]. Similarly, Sri Lankan Tamils have the option to marry under the Tamil cultural law, namely *Thesawalamai law* or the 1907 GMRO [23]. In the case of Muslims, Sri Lankan General Marriage Ordinance (GMO) prohibits Muslim marriages from taking place under GMRO. It provides the legality of marriage for Muslims when it takes place under the Muslim Marriage and Divorce Act of 1951 (MMDA) [24].

Hence, the marriage law that is applicable to Muslims in Sri Lanka is contained in the Muslim Marriage and Divorce Act of 1951 (MMDA). According to this act, the validity of a Muslim marriage

does not depend on "registration or non-registration"; rather, it is determined according to Islamic principles that are prescribed in MMDA [25].

The Muslim Marriage and Divorce Act of 1951 (MMDA) is administrated by the *Quazi* Court system [26], and presently there are 65 *Quazi* courts operating island-wide [27]. The Board of *Quazi* under the Ministry of Justice appoints *Quazi*. According to MMDA Section 12 (1), the Judicial Service Commission of Sri Lanka appoints a male Muslim of good character and honour as *Quazi* [28]. Hence, *Quazi*' appointment in Sri Lanka takes place on the basis of gender, honour, respect and religiosity of a person rather than academic and professional qualifications [29].

Several community activists have concerns about the quality of the service and practices of the *Quazi* Courts, while some Muslim women activists question the provision of a compulsory *Wali* (male guardian) for Muslim women in the MMDA [26]. They compare the MMDA to conventional law and describe these provisions as contradictory to the Sri Lankan constitution which allows Sri Lankan adults (male and female) to marry anyone on their own without the consent of a guardian [26]. They also accuse the MMDA of allowing child marriage [26]. Further, women activists claim that the MMDA provides a basis for inequality in terms of the consent of brides, conditions for divorce and an arbitrary nature of maintenance and compensation for different types of divorce and polygamy [26]. Women activists also consider preventing women from the appointment as *Quazi*s, marriage registrars, adjudicators or members of the Board of *Quazi*s as discrimination based on gender [26].

The Sri Lankan government has taken steps to address these issues and set up three national level committees since the 1970s [2]. The most recent is the Muslim Personal Law (MPL) Reforms Committee which was established in 2009 by the then Minister of Justice Mr. Milinda Moragoda under the heading of the retired Supreme Court judge Saleem Marsoof [2]. This team consisted of 16 experts on Muslim personal law including women as well as two representatives from ACJU.

In order to support this commission, the National Shoora Council of Sri Lanka (NSC-SL)—national-level Muslim religious leaders and a professional forum—invited experts in Muslim Family Law from Malaysia and organized a symposium under the title "Contemporary thoughts on Muslim Marriage & Divorce Law" in January 2016 [27]. The local and international experts presented papers and several Muslim intellectuals and representatives of NGOs including ACJU participated in this symposium. This symposium categorized the appointment of female *Quazi*, marriage age, *Wali* (consent from a guardian) and polygamy as *Shariah* matters and other issues including irregularities in *Quazi* courts as administrative problems.

Since ACJU is also a part of the MMDA reform committee, it worked with the MMDA team and provided necessary support and suggestions to review the required areas. However, ACJU broke away from the MMDA reform committee, disputing certain issues including women *Quazi* [30].

The objection of ACJU to women *Quazi* was founded on the basis of certain *Nas* (Verses of the Quran and Hadiths) that were commonly quoted by opponents of women judges (*Qudath*) as discussed above. Further, the anticipated methodology adopted by Muslim women activists, particularly their critiques of the compatibility of the Islamic legal system for present conditions [26], created suspicion of their motivations [21,31]. Hence, the issue of women *Quazi* in Sri Lanka became a matter of kudos between ACJU and women activists.

ACJU totally rejected the proposal to reform the Act and to allow the appointment of Quazi on the basis of qualification, ignoring the gender differences, even though they realized some irregularities in the present *Quazi* court system due to unqualified male Quazis. In the meantime, some women activists insist on changing the entire Quazi court system even though they found some indiscretions in the present court system due to some Quazis' inadequate administrative skills rather than their gender affiliation.

Henceforth, this research explores the variations of the gender debate through history, focusing on the changes of context and socio-political conditions as well as women's status. Moreover, this study constructs a view on women *Quazi* in Sri Lanka by articulating the perception of this issue in Muslim

countries, particularly in the countries that adopted *Shaifie Mazhab*—the *Shaifie* school of law in dealing with religious practices, such as Malaysia and Indonesia.

## 3. Research Method and Focus

### 3.1. Theoretical Foundation

The *Quranic* interpretation tradition in Islam is a well-developed science and is still expanding. It is known as the sciences of the *Quran* (*Ulum al-Quran*) and the basic principle of exegesis (*Usul alTafsir*). Since the *Quran* revealed to the Prophet Mohammad (pbuh) for 23 years (610–632 AD) to transform a pagan community into a civilized society, the verses of the *Quran* contain teachings and guidance addressing the issues in a particular context of a society and nature of people. *Ulum al-Quran* as a science deals with the nature of the text in a particular context, order of revelation and explanations of the types of expression used in the revelation as well as the method of recitation in order to bring out the meaning and application of this text for present context. Jalaluddin al-Suyuti (d.911/1505) provides a detailed discussion of this subject [32]. *Usul al-Tafsir* is an updated version of *Ulum al-Quran*. It contains detailed information on the history of the *Quran*, its structure and linguistic qualities and dynamics of the methodology of exegesis including historical information under the topic of *Makki* and *Madani*, *Asbab al-Nuzul* (occasion of revelation), *Muhkam Mutashabih* (clear and ambiguous verses) and *Nasikh Mansukh* (abrogating and abrogated verses), topics of *Ijaz* (inimitable) [33].

The knowledge on socio-political and historical context is important among all other aspects of *Usul al-Tafsir* to understand the text of the Quran and its meanings. Sattar (1977) studied the socio-political condition of the text and the importance of interpreting the text of the Quran in the classical Arabic literature and highlighted the dynamics of the text in different historical conditions [34]. Recep Dogan has given a comprehensive discussion on the significant role of history in exegesis of the *Quran*, referring to the works of Zurqani, Subhi Salih, Dhahabi and Manna' Qattan [35].

Thus, this research focuses on *Asbab al-Nuzul*—"occasion of revelation", a historical aspect of *Ulum al-Quran*. Many interpreters of the *Quran* in the 20th century have focused on *Asbab al-Nuzul* and approached this from different theoretical backgrounds. Aiysah Abdul Rahman, who was popular with Bint al-Shati (d.1998), for example, elucidated and expanded *Asbab al-Nuzul* by introducing a holistic, intra-textual, thematic and literary style of interpretation [36]. She understands Asbab al-Nuzul as referring to no more than the situation relating to specific passages of the *Quran*, upholding the famous principle of the Muslim jurists that the decisive factor (in determining the meaning of the verse) is the universality of the wording and not its specific cause [36].

Fadel, M. (2011) is another scholar who focused on *Asbab al-Nuzul* in recent times to understand the patterns of Islamic legal reforms. He identified two aspects of historicism that was relevant to his research: a progressive theory of history and history as a source for textual interpretation. He preferred history as a source for textual interpretation as it supports Islamic jurisprudential concept *Takhsis al-'Amm*—specification of the general term. It means the principles that general terms of the text must not be applied generally until care has been taken to exclude the possibility that circumstantial evidence indicates a more specific intent [28]. He explored this reform strategy by studying literary history, Islamic legal hermeneutics and substantive Islamic law. He demonstrated substantial egalitarian reform without fundamental changes to traditional Islamic theological doctrines [21].

### 3.2. Conceptual Framework

Thus, this research constructed the review framework referring to *Asbab al-Nuzul*—"occasion of revelation"—particularly focusing on history as a source of interpretation of texts. This means that the *Ayat* (verse of the *Quran*) contains a divine intent. This divine intent is universal and common for all communities and all times. However, it has been communicated to certain communities within their socio-political, economic and cultural context in order to make its immediate response to a specific condition. Hence, the interpreter of the *Ayat* has to understand the divine intent of a particular

text, extract this divine intent from the nexus of socio-political, economic and cultural contexts of the community that was revealed and to explain it to the present socio-political, economic and cultural context. This methodology is called "contextualization". This means "extracting the Divine message or Divine intent in a particular verse from the nexus of socio-economic, political and cultural contexts of revelation as well as the context of interpreters" [20]. Hence, the following diagram explains the process of the formation of the framework and the method of reviewing the interpretation in regard to women *Quazi*.

Although there is no *Nas* (Verses of the Quran or Hadith) that directly communicates on the subject of women judges (*Qudath*), the classical and modern scholars use some *Nas* that discusses women leadership to decide whether women can be appointed to leadership including in the legal sector. They understood these *Nas* in their contexts; some of them excluded women from leadership positions including the position of a judge (*Qadi*) while others included them and approved women judges (*Qudath*). This study, therefore, uses the contextualization method to review both groups'—the group that permits women judges (*Qudath*) and the group that opposes women judges (*Qudath*)—understanding of these *Nas* and to explain the subject of women judges (*Qudath*) in today's context. Further, the appointment of women *Quazi* to administrate a provision in Muslim personal law in Sri Lanka will be discussed based on the finding of women judges (*Qudath*). Particularly, this research will discuss whether the classical discussion on women judges (*Qudath*) can be taken as an analogy to decide the appointment of female *Quazi* in Sri Lanka.

Figure 1 explains the possible variations of context and its outcomes of a *Nas*. Due to no direct *Nas* on the subject of women judges (*Qudath*), the scholars constructed positions on the subject based on their understanding of some *Nas* related to women leadership within their contexts. Hence, the scholars have debated whether gender is a qualification of a judge (*Qadi*) within the Arab context and different socio-political and historical conditions. In the pre-modern period, the social condition of women including education was very minimum, particularly the condition of women in the Arab world was vulnerable [36]. The judicial area in the pre-modern Arab world was vast and only few judges (*Qadis*) managed the legal issues of the entire Arab world. The law during that time was not coded and the judge (*Qadi*) was required to deduce the law from the body of legal scholarship and literature. Hence, *Ijtihad*—the ability to deduce the law from the primary sources and body of legal scholarship—was an important qualification of a judge (*Qadi*), as discussed above. The judge (*Qadi*) in the Arab world during that time was considered equal to a ruler. The scholars discussed whether gender is a qualification of a judge (*Qadi*) or not in this context. However, the Sri Lankan context is different from theirs in term of socio-political and historical conditions as well as women educational qualifications. Therefore, Sri Lankan Muslims are required to discuss the issue based on their own context.

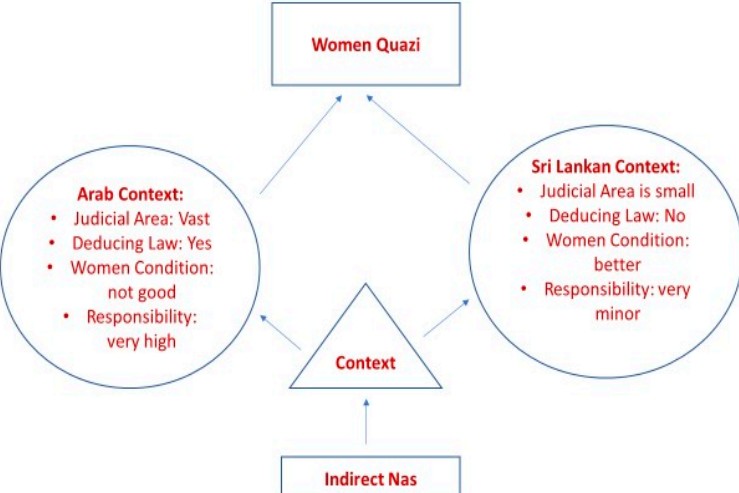

**Figure 1.** Conceptual framework of contextualization.

## 4. Findings and Discussion

This research explores the conceptual interpretation of *Quazi* who could administrate the legal dignitaries related to the marriage and divorce of Muslims in Sri Lanka. Although many Muslim women have worked as magistrates and judges in Sri Lankan conventional courts since 1970, objection for female *Quazi* has been put forward by ACJU as it is considered religious duty rather than legal administration. Hence, this study investigates whether gender is a qualification for the appointment of *Quazi*. Hence, the findings from the data can be categorized under five themes: women's condition, the role of *Quazi* in terms of the judiciary area, the role of *Quazi* in terms of responsibility, the role of *Quazi* in developing law and ensuring fair hearings and women's privacy.

### 4.1. Women's Condition

This research found that the condition of women during the time of revelation of the *Quran* as well as the pre-modern Arab world is different from the condition today. Women's conditions in terms of their academic and social contribution during the early period of Islam were pitiful, although Islam provided enough motivation for education and social contributions [37]. Since there were limited opportunities for women to have exposure to intellectual communities, one hardly found qualified women to be appointed to leadership positions, including the post of a judge (*Qadi*) [38]. However, women who were qualified enough were appointed for various positions and they played significant roles. For instance, the Prophet (pbuh) appointed Samra binti Nuhaik Al-Asadiyyah as the administrator (*Muhtasib*) of commercial law [39], where the violation of rights and the potential of transgression in dealings is high. This shows that the Prophet (pbuh) saw the qualifications of a person rather than their gender in appointing the judge (*Qadi*).

Further, this report continued that Samra binti Nuhaik was in the same position during subsequent rulings of the first two caliphs: Abu Bakar (R.A) and Umar (R.A) [40]. She was not only allowed to continue this job during their terms; rather, they encouraged her to preside the position by requesting her to provide reports on the market's situation [39]. Ibn Al-Jawzai said that Caliph Umar, upon entering the market, used to visit Samra binti Nuhaik for an update on market issues [38]. This conveyed the idea that Caliphs including Umar, the strict personality among the Prophet's companions on women's affairs (Al-Bukhari, Hadith No:68), did not see gender as a factor that prevented women from the appointment of a judge (*Qadi*) if she was qualified enough.

Similarly, Caliph Umar appointed another woman, Al-Shifa Binti 'Abdullah (Laila), to administrate commercial law [41]. As she was well known for her knowledge, piety, morality and respect, she was appointed to this post [42]. It should be noted that the post of *Hisbah*, market inspector, during that time was a highly respectable job and there were enough qualified men. Hence, the appointment of Al-Shifa (Laila) to this post clearly delivers the message that Caliph Umar did not see femininity as a disqualification to the post of administrating the law.

Hence, these incidences show that gender was not considered as a qualification in the appointment of judges (*Qudath*); rather, the prophet Mohammad (pbuh) and his companions deliberated on educational and other qualifications of a person to the post of a judge (*Qadi*) to administrate the law. Even though these incidences need further investigation regarding their authenticity [43], the meaning is consistent with the history, particularly the prominent place of Aiysah (R.A) and other women in different social leadership.

As women's education becomes irrepressibly better today [44], the performance of girls in schools and women in higher learning institutions is increasing [45], allowing women proceeding into the courts to become compulsory.

### 4.2. The Role of Quazi in the Context of a Judicial Area

The analysis of texts shows that traditional scholars discussed the issue of women judges (*Qudath*) when the territory of Muslim administration was vast, extending from Europe to Russia and China

to India. In this vast area, there were only a few courts [46]. Therefore, classical scholars might have considered gender as an important qualification for judges (*Qudath*) as the post requires travelling frequently to far distances alone to investigate cases. However, the situation has changed now, and the judicial boundary of a court has become constricted. The number of courts has also increased. Therefore, the context of the discussion of classical scholars is different from ours.

### 4.3. The Role of Quazi in Terms of Responsibility

The findings also revealed that classical scholars, including four great *Imams* in Islamic jurisprudence, debated the qualifications of a judge (*Qadi*) in the context of administrating an entire judiciary including commercial, criminal and administrative as well as other legal sections. Particularly during the period of four imams, namely Abu Hanifa al-Nu'man Bin Thabith (AD 699–767), Malik Bin A*nas* (AD 715–796), Mohammad Bin Idrees al-Shafie (AD 766–820) and Ahmad Bin Hanbal (AD 780–855), the judges (*Qudath*) were the people who were in charge of administrating the entire judiciary [47]. Zubaida summarized the judge (*Qadi*)'s duties during this period from the legacy of Mawardi (AD 972–1058) and Ibn Khaldun (AD 1332–1406) [48]. She listed the duties of a judge (*Qadi*) as "Judgement in litigation, the protection of the incapable, the administration of *Awqaf*—mortmain charitable endowment property, the execution of wills and testaments, the guardianship of unmarried women without guardians, the application of *Hudud*—criminal punishments as stipulated in the *Quran*, the policing of public buildings and roads and the control of *Shuhud* (Singular-Shahid)—eye witness and other court auxiliaries and the appointment of deputy judges (*Qadis*) [48]. Ibn Khaldun adds to this list the *Mazalim*-grievances, usually a function of the ruler, the conduct of jihad and control of the mint." [48] According to this, the judge (*Qadi*) during those periods presided over many ministries in today's context. Furthermore, these scholars considered the judge (*Qadi*) as a delegation of the power of the political leader, making decisions on behalf of him in critical situations [49]. However, the situation of Sri Lankan Muslims is totally different from this context. The classical scholars discussed the qualification of judges (*Qudath*) to administrate the entire shariah law with the abovementioned duties, while Sri Lankan Muslims discuss the qualification of *Quazi* who can administrate a provision in the civil law of Sri Lanka.

### 4.4. Role of Quazi in Developing Law

This study noticed that deducing the law from the primary sources of Islam was the primary role of a judge (*Qadi*) during the periods of classical scholars. Martin La described the function of a judge (*Qadi*) during those periods and said that the judge (*Qadi*) not only administrated the body of law during these periods, however they also developed law from the primary sources [50]. Thus, the classical scholars discussed *ijtihad*—the ability to process legal decisions independently on the basis of the primary sources of Islam, *Adalah*—the genuineness of a judge in their sayings, doings and jurisdictions and *Kamal*—mental and physical perfection as primary qualifications of a judge (*Qadi*). In that perspective, many of the traditional scholars disqualified women from the office of judge (*Qadi*) due to their qualifications [51]. However, the Muslims of Sri Lanka do not discuss the appointment of women *Quazi* in this context; rather, they are discussing qualified women rising to the post of *Quazi* to administrate family disputes in the purview of the Muslim personal law Act.

### 4.5. Ensuring Fair Hearings and Women's Privacy

The research also found concern regarding fair hearings in classical Islamic judiciary systems [52]. In order to ensure fair hearings, the Prophet Mohammad (pbuh) appointed his wives to listen to the women who sought solutions in family disputes. For instance, the wife of Abdur Rahman bin Al Zubair Al-Qurazi (R.A) came to Aisha (R.A) wearing a green veil and accused her husband of domestic violence. She showed her a green spot on her skin that was caused by the beating. Aisha (R.A) counselled her and requested her to wait for the Prophet (pbuh). Upon the arrival of the Prophet (pbuh), Aisha (R.A) explained the grief of Abdur Rahman's wife and said, "I have not seen any woman

suffering as much as the believing women. Look! Her skin is greener than her clothes". The Prophet (pbuh), having heard the incident from Aisha (R.A), confirmed the domestic violence between Abdur Rahman and his wife and gave her a fair solution (Al-Bukhari, Hadith No:5825). This incident shows that the Prophet Mohammad (pbuh) appointed his wives to listening to women who accused their husbands of domestic violence and issues with their intimate relationship.

Another incident strengthens the prophet's method to ensure proper hearing. One day, a young girl came to Aisha (R.A) with great sorrow and said that her father wanted her to marry his cousin even though she did not want to. Aisha (R.A) listened to her and asked her to wait until the Prophet (pbuh) returned home. When the Prophet (pbuh) came home, Aisha (R.A) explained the girl's situation to him. The Prophet (pbuh) then sent a message to the girl's father, calling him to come to his home. Upon the arrival of the girl's father, the Prophet (pbuh) asked him about his daughter's complaint relating to her marriage. After listening to her father, the Prophet (pbuh) said to her, "Accept what your father has arranged". She responded, "I do not wish to accept what my father has arranged". Then, the Prophet (pbuh) said, "this marriage is invalid and marry whomever you wish". She said, "I have accepted what my father has arranged, however I wanted women to know that fathers have no right in their daughter's matters i.e. "they have no right to force a marriage on them" (Al Nasa'i, Hadith No:3269). In this incident, Aisha (R.A) received the girl and listened to her, then informed the Prophet (pbuh) about her worries too. This event also reveals that the Prophet (pbuh) appointed his wife Aisha (R.A) to listen to women who complain of their immediate family members on transgressing their rights and women affairs.

In both of these occasions, although the Prophet (pbuh) could have listened to them directly, he preferred his wives to listen to the women considering fair hearing and women's privacy. This is because women like to open up to other women and to discuss the issues related to family disputes, particularly complaints against their immediate family members.

Maryam Azwer (2018), for example, highlighted the absence of a woman in the *Quazi* court system in Sri Lanka and the failure of ensuring fair hearings [52]. She studied the case of a woman named Zainab who applied for a divorce from her abusive husband and waited for years to get a solution as she could not discuss with a male *Quazi* about her husband's physical and psychological tortures [53]. The Women's Action Network documented similar abusive cases of women by their husbands from the northern and eastern region of Sri Lanka [54]. Shreen Abdul Shakoor said that half of the women she interviewed on their experience in Sri Lanka *Quazi* courts were highly concerned regarding fair hearings and women's privacy. This is because the women were not only required to describe the issues relating to intimate relationships to a male *Quazi*, however they also had to do it in an open environment where others could listen to them. She continued by saying that some women have described such situations as relating to "private issues" that cannot be discussed with male *Quazi*. Shreen Abdul Shakoor further said that some women were afraid of being exploited sexually later. Hence, she demands women *Quazi* in order to ensure fair hearings [55].

Since ensuring fair hearings is essential in Islamic judiciary including marriage and divorce law [52], the Prophet (pbuh) appointed his wives such as Aisha (R.A) to listen to women who made complaints against their husbands as well as family disputes. Hence, the appointment of women *Quazi* in the present *Quazi* court system in Sri Lanka might be a way of ensuring this essential element in court hearings.

The findings above not only show that Muslim scholars in the past debated the qualifications of judges (*Qudath*) considering their socio-cultural and political context, however they also highlight the necessity of revising these qualifications including their argument on gender differences in relation to today's context. Further, these findings expose that some scholars in the past prohibited women from the post of a judge (*Qadi*) considering their context, particularly women's education level, exposure and social status, while their contemporaries in some other areas or their disciples later in the same area allowed women to proceed into the courts when they found them eligible and educated. Hence,

the appointment of female judges (*Qudath*) is a local matter and the local socio-cultural context should be considered to decide whether women are eligible to proceed into the court or not.

Shiekh Yosuf al-Qaradawi [56] and the religious institution of Egypt's government [55] as well as the Malaysian religious affairs ministry [56] also endorse it and found the issue of appointing women judges (*Qudath*) to be a local issue that could be determined by a local body considering the local context and requirements [57].

As far as there is no specific *Nas* (the verse of the *Quran* or Hadith) on the appointment of female judges (*Qudath*) and the appointment of female judges (*Qudath*) is a local matter, Muslims of Sri Lanka also have to consider the appointment of women *Quazi* based on the Sri Lankan context and the requirements. The socio-cultural context of Sri Lanka is different from the Arab world, particularly during the classical scholars' period, particularly relating to women's education, the court system and the duty of *Quazi*. Women in our context are not only educated and qualified, rather they are excelling in many fields of study including legal affairs. The ratio of women's enrolment in higher learning institutions in Sri Lanka is higher than men. Women's social status has also improved in Sri Lanka compared to their status in many Arab countries. Meanwhile, the duty of *Quazi* in Sri Lanka is very minor in association with the duties of judges (*Qudath*) during the classical scholars' period. The judicial area of *Quazi* in Sri Lanka is also narrow compared to the judicial area of judges (*Qudath*) in the pre-modern Arab world. Hence, the socio-political and cultural context of Sri Lankan Muslims today is different from the context of Arabs, particularly in the context of scholars who discussed the qualification of judges (*Qudath*) in the pre-modern Arab world.

Therefore, the discussion among the scholars on the qualification of judges (*Qudath*) with special reference to gender seems irrelevant to Sri Lankan Muslims' discussion on female *Quazi*. Further, women face many difficulties in the present *Quazi* court system in Sri Lanka not because of unqualified male *Quazi* alone, however also due to the absence of women *Quazi* who can listen to them and understand their issues.

Thus, women who are qualified enough could be considered to be appointed as *Quazi* when the society requires their service in Sri Lanka. As per needs and social change, many Muslim countries have considered this suggestion and appointed women judges (*Qudath*) in *Shariah* courts, including Malaysia, as discussed above [58].

## 5. Conclusions

In the absence of a precise direction in the *Quran* and Hadith on the qualification of a judge (*Qadi*), the female judge (*Qadi*) has been a matter of discussion among Muslim scholars throughout history. The differences of opinion among them on qualifications have been summarized in this article and the dispute among them regarding gender has been shown.

Even though some of them opposed the appointment of women judges (*Qudath*) on the basis of gender, others approved women's appointment for the post of a judge (*Qadi*) on the basis of qualifications. This shows that the scholars opined on women judges (*Qudath*) based on their understanding of certain *Quran*ic verses as well as Hadiths related to women leadership in their contexts. Hence, considering the context and needs, the Muslim countries appointed women judges (*Qudath*) to administrate the Shariah law. Thus, the prohibition of the appointment of women judges (*Qudath*) is not *Ijma*—consensus or agreement of the Muslim scholars—as claimed by some scholars, whereas a considerable number of main-stream Muslim scholars have different opinions on the issue.

Hence, this research found the discussion on the qualifications of a judge (*Qadi*) among the classical scholars to be extraneous and irrelevant to the discussion on appointing women *Quazi* in Sri Lanka as their context is different from the context of Muslims in Sri Lanka. In this perspective, it is inappropriate to prohibit the appointment of women *Quazi* in Sri Lanka based on classical scholars' discussion on the qualifications of a judge (*Qadi*).

Further, women have been appointed to many leadership posts, including judiciary as judges (*Qudath*), to administrate Shariah Law during the early period of Islam as well as throughout history.

In the present world, many Arab and Muslim countries including Malaysia, Indonesia, Egypt and Tunisia appointed women as judges (*Qudath*) to the *Shariah* court. According to Ramizah Wan Muhammad (2005), for instance, the appointment of women *Quazi* in Malaysia was endorsed by the provisions of the Administration of Islamic Law (Selangor) Enactment 2003, Section 56 and 58 [58,59].

It should be noted that some of these countries, including Malaysia and Indonesia, official *Mazhab*—school of Islamic law is *Shafie Mazhab*—*Shafie* School of Islamic law, which is the same as the *Mazhab*—school of Islamic law of Sri Lankan Muslims. The present Muslim marriage and divorce law of Sri Lanka, which was indeed originally exported from Indonesia (Batavia) by the Dutch Governor Falck in 1701 [60] as Indonesia's official *Mazhab*—school of Islamic law was *Shafie Mazhab*.

Moreover, bringing the classical scholars' discussion to bear on the qualification of judges (*Qudath*) and insisting these qualifications for the appointment of *Quazi* in Sri Lanka excludes not only women, however also the majority of male *Quazi*s. This is because many classical scholars, as discussed before, prescribed *Ijtihad*—the ability to process legal decisions independently on the basis of the primary sources of Islam as a basic qualification for a judge (*Qadi*). However, there is no qualified person among Sri Lankan Muslims who could exercise *Ijtihad* to be appointed as *Quazi* in the present time. Interestingly, there is no record of such *Mujtahid*—the person who could exercise *Ijtihad*—in Sri Lankan Muslim history. Unfortunately, the majority of these *Quazi*s hardly have knowledge to access the original Arabic texts of Muslim personal law.

Hence, this article recommends the appointment of female *Quazi* in Sri Lanka if women are qualified enough in terms of academic and other requirements. Further, this article supports female *Quazi* in the present *Quazi* court system of Sri Lanka, considering the privacy of women as well as *Maslaha*—social concerns.

**Author Contributions:** Conceptualization, Methodology, Formal Analysis, Writing-Original Draft Preparation and Writing-Review & Editing, M.I.R., and Supervision as well as validation of methodology were done by S.G.

**Funding:** This research received no external funding.

**Conflicts of Interest:** The authors declare no conflict of interest.

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
