# Peer review of "Women Quazi in a Minority Context: An Overview of Sri Lankan Experience"

_societies, doi:10.3390/soc9010013_

Round 1
Reviewer 1 Report
This piece addresses interesting subject matter and has scholarly potential. For this potential to be developed, the piece requires significant substantive and stylistic revisions.
The reader needs more basic definitions & translations for key terms (e.g. Sharia, ACJU, Adalah, etc.,), and brief basic background for non-experts in Sharia law (vs. for e.g. Common law), and its aspects of interpretation— for e.g. most of p.7 will be largely meaningless to non-experts.
There are also throughout the piece a significant number of sentences & passages that are unclear, vague, or need further explanation (many are highlighted by the reviewer in the text). The author’s arguments are clearest in their conclusion: they should look there for guidance in revision. There are also sections where the evidence does not appear to match the claim or is weak (e.g. 1st & 2nd para. In section 4.5, top of p.9). The piece requires stronger proof-reading of grammar, including re: ESL/translation, and the reviewer suggests moving Figure 1 below the discussion that explains it — in its present placement its meaning is largely unclear.
Most importantly, the author’s gender analysis requires significant clarification & elaboration.
With regard to the analysis the author needs to further explain the dynamics of gendered discrimination & power dynamics in the following sorts of ways:
– re: the basic gender <> judicial practice arguments, “Hanafi School allowed women Quazi to administrate all laws except Hudud—criminal cases—and Qisas—retaliation cases” — why not these types of law? The author focuses on “women Quazi to administrate the Muslim personal law—legal dignitaries related to marriage and divorce as well as inheritance—in Sri Lanka” — why these types of law & not others?
— “According to MMDA Section 12 (1), this appointment takes place on the basis of the honour, respect and religiosity of a person rather than qualifications [27]. Hence, so far, this is strictly reserved for men.” What do these sentences mean?
— In the section on women being appointed judge “without considering her femininity” is it not possible that their femininity was considered, and they were appointed judge? Relatedly, when the author writes “gender as a qualification” do they actually mean masculine gender? Further relatedly (and centrally), the author’s arguments re: gender & qualification for the judiciary contradict each other — they argue at points that women Quazi should be used because women can speak more freely to them about domestic violence (because they too are women), but then conclude that women Quazi should be appointed “ignoring gender differences”
— The author uses the term “properly” repeatedly as an unclear shorthand (which appears to relate to their arguments about gender & law) — they need to spell out what exactly they mean in each context.
— Also centrally, the issues around fundamentalism / literalist interpretations of the Quran as they relate to gender (& other vectors of power) needs to be addressed substantively — are all sections in the Quran interpreted literally, or is this literalism selective in ways that reflect the unequal distribution of power? If fundamentalism is selective, is gender the only vector of power reflected in that selectivity (is, for e.g., class bias, homophobia, etc., also part of these dynamics)? The author also writes that “Islam prescribes women to cover” and cites 1 section in the Quran—again, are all sections of the Quran followed literally, or just some (and reflecting unequal power)? This is particularly relevant / important where alternate explanations are possible for the author’s claims. For e.g., where they write “As far as Islam prescribes…Muslim women in Sri Lanka do not like….” There is tenuous evidence for their claim, and arguably other (socio-political) explanations can be provided for what these women do and do not like. — Finally, what does the author intend by framing wives reporting their husbands’ abuse as “complaining”?

Author Response
Dear Reviewer
Thank you very much for reviewing our manuscript and for giving us useful suggestions. Your suggestions not only help us to expand the article but also broaden our perspective. We have tried our best to incorporate your suggestions and to revise the manuscript.
Thank you

Reviewer 2 Report
The essence of the argument, as I read it, is that the Islamic traditions (in theory and practice alike) were less restrictive of women's potential leadership, especially in juridical and judicial activities, than some modern Muslims take these traditions to be doing. This may be stated more clearly and perhaps repeated. The alleged broad ban on women's judicial appointments is attacked effectively and weakened by the end of the article, but it can more assertively be declared irrelevant to normal functions of everyday life. The text supposedly says that women should not be entitled "the people's affairs" (ما أفلح قوم ولوا أمرهم امرأة). This is vague and unproductive. A woman doctor is someone with power and has entitlements over the people's affairs, and no one could argue that women's practice of medicine is problematic. However, the legal tradition does have strange emphases on women's domestic roles, and this may simply be explained as a function of the idealistic nature of juristic writing in their short format. Long legal commentaries have sufficient caveats about women's presence in society and provide sufficient data that go against the supposed broad ban. It is only that many of these texts remain in manuscript form, and curiosity about them is limited.
A crucial moment on page 5 was an opportunity to explain how divine texts 'brevity' ought to force interpreters to handle them with care:
This divine intent is universal and common for all communities and all times. However, it has been communicated to certain communities within their socio-political, economic and cultural context in order to make its immediate implementation easy to understand. Hence, the interpreter of the Ayat has to understand the divine intent of a particular text, extract this divine intent from the nexus of socio-political, economic and cultural contexts and explain it to the present socio-political, economic and cultural context. End of quote.
This may need some development and clarification to show that an interpreter who takes advantage of the brevity of a text to impose a ready view is not in a better position than any other reader with a different view. I could take the same instruction on the protection of women to mean that their presence in society requires more effort to facilitate and normalize their interaction with men, instead of forcing them to stay home. Again, there is no 'precedence' power that leads to the conclusion of a blanket domestication of women. Historically, women's participation in society was comprehensive in scope, attested by sources of administrative history, which scholars who defended women's role as 'family-bound' have not examined.
Some limited revisions, aiming at producing smoothness for the reader would also be a good idea to push the piece closer to the publication stage.
Author Response

(The authors gave the same response as above.)
